# Sharp Recovery Thresholds of Tensor PCA Spectral Algorithms

**David L. Donoho**
Department of Statistics
Stanford University
donoho@stanford.edu

**Michael J. Feldman**
Department of Statistics
Stanford University
feldman6@stanford.edu

## Abstract

Many applications seek to recover low-rank approximations of noisy tensor data. We consider several practical and effective *matricization* strategies which construct specific matrices from such tensors and then apply spectral methods; the strategies include tensor unfolding, *partial tracing*, power iteration, and recursive unfolding. We settle the behaviors of unfolding and partial tracing, identifying sharp thresholds in signal-to-noise ratio above which the signal is partially recovered. In particular, we extend previous results to a much larger class of tensor shapes where axis lengths may be different. For power iteration and recursive unfolding, we prove that under conditions where previous algorithms partially recovery the signal, these methods achieve (asymptotically) exact recovery. Our analysis deploys random matrix theory to obtain sharp thresholds which elude perturbation and concentration bounds. Specifically, we rely upon recent *disproportionate* random matrix results, which describe sequences of matrices with diverging aspect ratio.

## 1   Introduction

Tensors—multi-way arrays—and tensor methods are fundamental to modern data analysis. Data given by three or more indices are increasingly the focus in diverse areas including signal and image processing [9, 19, 22], while high-order moments represented by tensors are of interest in problems such as community detection and learning latent variable models [1, 2, 8].

The *spiked tensor model*, introduced by Montanari and Richard [20], is a simple statistical model of tensor data with latent low-rank structure. Observed data $X$ of dimensions $n_1 \times n_2 \times \cdots \times n_k$ are the sum of a low-rank tensor and noise:

$$X_{i_1,i_2,\ldots,i_k} = \lambda v_{1,i_1} v_{2,i_2} \cdots v_{k,i_k} + Z_{i_1,i_2,\ldots,i_k}, \qquad j \in [k],\, i_j \in [n_j], \qquad (1)$$

where $k \geq 2$ is the tensor order, $\lambda$ is a signal strength, $v_j \in \mathbb{S}^{n_j-1}$, $j \in [k]$, are unit vectors, and $Z$ is a noise tensor. Noise entries are assumed to be independent standard Gaussians:

$$Z_{i_1,i_2,\ldots,i_k} \overset{i.i.d}{\sim} \mathcal{N}(0,1), \qquad j \in [k],\, i_j \in [n_j].$$

Estimation of $v_1,\ldots,v_k$ and the low-rank component of $X$ generalizes to tensors the problem of low-rank matrix approximation—principal component analysis (PCA)—and is therefore known as tensor PCA.

This model reduces for $k = 2$ to the *spiked matrix model*, with maximum likelihood estimators of $v_1$ and $v_2$ given by the first left and right singular vectors $\hat{v}_1$ and $\hat{v}_2$ of $X$, respectively. For $k \geq 3$, however, computation of the maximum likelihood estimators of the low rank component amounts to solving an NP hard problem (Hillar and Lim [14]),

$$\sup_{j \in [k],\, u_j \in \mathbb{S}^{j-1}} \sum_{i_1,\ldots,i_k} X_{i_1,i_2,\ldots,i_k} u_{1,i_1} \cdots u_{k,i_k}, \qquad (2)$$

37th Conference on Neural Information Processing Systems (NeurIPS 2023).

necessitating alternate, efficient algorithms.

## 1.1 Contributions

This paper studies *matricization*-based approaches, which convert tensors to matrices (by reshaping or by contracting constructions to be described) and then apply spectral methods. We assume the spiked tensor model in a high-dimensional asymptotic framework in which each array dimension $n_1, \ldots, n_k$ tends to infinity; we make no assumptions on their relative rates. While previous analyses of tensor PCA algorithms have assumed order $k = 3$, supersymmetry $v_1 = v_2 = \cdots = v_k$, or hypercubical format $n_1 = n_2 = \cdots = n_k$, we permit tensors of diverse dimensions $n_j$ and unrelated $v_j$.

In this setting, we fully analyze tensor unfolding or reshaping, a widespread technique [12, 19]. Additionally, we analyze the *partial tracing* approach of Hopkins et al. [16], discovering its asymptotic equivalence to unfolding in performance—an improvement over previous analysis. We identify sharp thresholds in signal-to-noise ratio above which these algorithms partially recover the signal, and provide exact formulas for their limiting performance. Finally, we study generalizations of the power iteration and recursive unfolding algorithms of [20]. Above the same thresholds characterizing partial recovery of unfolding and partial tracing, these algorithms achieve (asymptotically) exact signal recovery. In other words, for signal-to-noise ratios such that unfolding or partial tracing partially recover the signal, power iteration and recursive unfolding exactly recover the signal.

Our approach relies upon fundamental and penetrating random matrix theoretic (RMT) results. Precise analysis of tensor PCA algorithms, including pinpointing of phase transitions and sharp performance quantification as provided here, is not possible purely via more standard tools in theoretical machine learning, such as matrix concentration and vector perturbation bounds. Specifically, we rely upon recent *disproportionate* random matrix results (introduced in Section 1.3), which describe sequences of matrices with diverging aspect ratio.

This work concisely demonstrates in the context of tensor PCA the advantages of an RMT-backed approach. Indeed, application of spiked matrix results yields simple and elegant theorems and proofs which are easily read. For example, the analysis of the partial tracing in [16], involving challenging matrix concentration and perturbation calculations, is reduced to a few careful lines. Though our results are admittedly purely asymptotic, there are no hidden constants or logarithmic factors. Even for modest-sized tensors, simulations demonstrate close agreement with theory.

## 1.2 Assumptions and Notation

Without loss of generality, we assume $\lambda \geq 0$ and $n_1 \leq n_2 \leq \cdots \leq n_k$, taking $n_1$ to be the "fundamental" problem index; that is $n_j = n_j(n_1)$, $j \in [k]$. We say an estimator $\hat{v}_j$ *partially recovers* $v_j$ if almost surely,

$$\liminf_{n_1 \to \infty} |\langle v_j, \hat{v}_j \rangle| > 0 \,,$$

and $\hat{v}_j$ *exactly recovers* $v_j$ if $|\langle v_j, \hat{v}_j \rangle| \xrightarrow{a.s.} 1$. In words, partial recovery demands a positive limit for the cosine similarity, while exact recovery requires asymptotically perfect cosine similarity.

We make the rank-one signal assumption in model (1) purely for expository efficiency; the spectral algorithms in this paper naturally generalize to rank-$r$ spiked tensors of the form

$$X_{i_1, i_2, \ldots, i_k} = \sum_{i=1}^{r} \lambda_r v_{1,i_1}^{(i)} v_{2,i_2}^{(i)} \cdots v_{k,i_k}^{(i)} + Z_{i_1, i_2, \ldots, i_k} \,, \qquad j \in [k], \, i_j \in [n_j] \qquad (3)$$

where $r \geq 1$ is the rank and $\{v_j^{(1)}, \ldots, v_j^{(r)}\} \subset \mathbb{S}^{n_j - 1}$ are orthonormal sets, $j \in [k]$. Such tensor decompositions are unique by Kruskal's theorem [18]. Where we estimate $v_j^{(1)}$ by the first right singular vector of an appropriate matrix $M_j$, $v_j^{(2)}, \ldots, v_j^{(r)}$ may be estimated by the subsequent right singular vectors, with analogous theoretical guarantees holding (that is, $v_j^{(i)}$ is the right singular vector associated with the $i$-th largest singular value of $M_j$).

We denote by $\otimes$ both the tensor outer product and Kronecker product (the latter is simply a vectorization of the former); it is clear from context which is meant. Let $\times_j$ denote multiplication between a tensor along the $j$-th axis and a vector of conformable dimension. The notation $a(n_1) \lesssim b(n_1)$

means $a(n_1) \leq Cb(n_1)$ for a constant $C > 0$ and all sufficiently large $n_1$, and $a(n_1) \asymp b(n_1)$ means $a(n_1) \lesssim b(n_1) \lesssim a(n_1)$.

Finally, we introduce tensor *slices*. Let $T \in \mathbb{R}^{n_1 \times \cdots \times n_k}$ and fix $\ell \in [k]$. The set $\{T_{i_1,\ldots,i_l} : j \in [l], i_j \in [n_j]\}$ contains the slices of $X$ of order $k - \ell$. Slices satisfy $T_{i_1,\ldots,i_\ell} \in \otimes_{j=\ell+1}^k \mathbb{R}^{n_j}$ and have entries

$$(T_{i_1,\ldots i_\ell})_{i_{l+1},\ldots,i_k} = T_{i_1,\ldots,i_k}, \qquad\qquad j \in \{\ell+1,\ldots,k\}, i_j \in [n_j].$$

## 1.3 Spiked Matrix Model

The behavior of matricization-based algorithms fundamentally derives from spectral properties of the spiked matrix model,

$$X_{i_1,i_2} = \lambda v_{1,i_1} v_{2,i_2} + Z_{i_1,i_2}, \qquad\qquad i_1 \in [n_1],\, i_2 \in [n_2]. \tag{4}$$

This model is extensively studied in random matrix theory, particularly under *proportional growth*, where $n_1$ and $n_2$ are of comparable magnitude:

$$n_1, n_2 \to \infty, \qquad \gamma_{n_1} = \frac{n_1}{n_2} \to \gamma > 0.$$

Under proportional growth, the sample covariance matrix $XX^\top/n_2$ does not converge to the identity (its expectation), and the leading left and right singular vectors $\hat{v}_1$ and $\hat{v}_2$ of $X$ are inconsistent estimators of $v_1$ and $v_2$, respectively. We highlight the following results of Benaych-Georges and Rao Nadakuditi [7], who establish formulas for the limiting misalignment of $\hat{v}_1$ and $\hat{v}_2$:

**Lemma 1.1.** *Let $\hat{v}_1$ and $\hat{v}_2$ denote the leading left and right singular vectors of $X$, respectively, and define*

$$c^2(\tau, \gamma) = \begin{cases} 1 - \dfrac{\gamma(1+\tau^2)}{\tau^2(\tau^2+\gamma)} & \tau > \gamma^{1/4} \\ 0 & \tau \leq \gamma^{1/4} \end{cases}.$$

*Under $\lambda = \tau(1 + o(1))\sqrt{n_2}$, where $\tau \geq 0$ is fixed, and $\gamma_{n_1} \to \gamma \in (0,1]$,*

$$|\langle v_1, \hat{v}_1 \rangle|^2 \xrightarrow{a.s.} c^2(\tau, \gamma), \qquad\qquad |\langle v_2, \hat{v}_2 \rangle|^2 \xrightarrow{a.s.} c^2(\tau\gamma^{-1/2}, \gamma^{-1}). \tag{5}$$

Recently, Ben Arous et al. [5] and Feldman [13] independently studied the spiked matrix model under *disproportional growth*, where $\gamma_{n_1} \to 0$ or $\gamma_{n_1} \to \infty$. Transposing $X$ in case $\gamma_{n_1} \to \infty$, we assume without loss of generality that $\gamma_{n_1} \to 0$. Phase transitions for the left and right singular vectors no longer coincide; $v_1$ is reliably estimated at signal strengths much weaker than $\lambda \gtrsim \sqrt{n_2}$. Drawing upon results in [5], [13], and [7], we have the following disproportionate analogs of Lemma 1.1:

**Lemma 1.2.** *Let $\hat{v}_1$ and $\hat{v}_2$ denote the leading left and right singular vectors of $X$, respectively, and define*

$$\overleftarrow{c}^2(\tau) = (1 - \tau^{-4})_+, \qquad\qquad \overrightarrow{c}^2(\tau) = \frac{\tau^2}{1+\tau^2}.$$

*Under $\lambda = \tau(1 + o(1))(n_1 n_2)^{1/4}$, where $\tau \geq 0$ is fixed, and $\gamma_{n_1} \to 0$,*

$$|\langle v_1, \hat{v}_1 \rangle|^2 \xrightarrow{a.s.} \overleftarrow{c}^2(\tau), \qquad\qquad |\langle v_2, \hat{v}_2 \rangle|^2 \xrightarrow{a.s.} 0, \tag{6}$$

*while under $\lambda = \tau(1 + o(1))\sqrt{n_2}$,*

$$|\langle v_1, \hat{v}_1 \rangle|^2 \xrightarrow{a.s.} 1, \qquad\qquad |\langle v_2, \hat{v}_2 \rangle|^2 \xrightarrow{a.s.} \overrightarrow{c}^2(\tau). \tag{7}$$

Limits (7) are corollaries of Theorems 2.9 and 2.10 of [7]. We note that [5] requires $n_2$ is polynomially bounded in $n_1$. This assumption, however, is necessary only to establish non-asymptotic bounds; the almost-sure results in Lemma 1.2 hold as $\gamma_{n_1} \to 0$ arbitrarily rapidly. While not explicitly stated in these references, it is easily verified that under proportional growth, $\liminf \lambda/\sqrt{n_2} > \gamma^{1/4}$ implies partial recovery of $v_1$ and $v_2$. Analogous statements hold for disproportional growth.

## 2 Tensor Unfolding

We study a general unfolding procedure that permits (1) tensors of general, unequal axis lengths and (2) unfolding along arbitrary sets of axes.

Let $N_j = \prod_{\ell=j}^{k} n_\ell$, $j \in [k]$, and for $S \subset [k]$, $S \neq \emptyset$, let $N(S) = \prod_{\ell \in S} n_\ell$. We define a map $\text{Mat}_S : \mathbb{R}^{n_1 \times \cdots \times n_k} \to \mathbb{R}^{(N_1/N(S)) \times N(S)}$ as follows: for indices $i_j \in [n_j], j \in [k]$,

$$a = 1 + \sum_{j \in [k] \setminus S} (i_j - 1) \prod_{\ell \in [k] \setminus S: \, j < \ell} n_\ell, \qquad b = 1 + \sum_{j \in S}(i_j - 1) \prod_{\ell \in S: \, j < \ell} n_\ell \qquad (8)$$

and

$$[\text{Mat}_S(T)]_{a,b} = T_{i_1, i_2, \ldots, i_k} . \qquad (9)$$

When unfolding along a single axis, we write $\text{Mat}_j = \text{Mat}_{\{j\}}$. In this case, (8) reduces to

$$a = 1 + \sum_{\ell=1}^{j-1}(i_\ell - 1)\frac{N_{\ell+1}}{n_j} + \sum_{\ell=j+1}^{k}(i_\ell - 1)N_{\ell+1} \qquad (10)$$

(taking $N_{k+1} = 1$) and $b = i_j$. The unfolded matrix $\text{Mat}_j(X)$ is a spiked matrix with aspect ratio $n_j^2/N_1$:

$$\text{Mat}_j(X) = \lambda(v_1 \otimes \cdots \otimes v_{j-1} \otimes v_{j+1} \otimes \cdots \otimes v_k)v_j^\top + \text{Mat}_j(Z) .$$

Similarly, $\text{Mat}_S(X)$ is a spiked matrix with aspect ratio $N(S)^2/N_1$:

$$\text{Mat}_S(X) = \lambda(\otimes_{j \in [k] \setminus S} v_j)(\otimes_{j \in S} v_j)^\top + \text{Mat}_S(Z) . \qquad (11)$$

We now review prior results on unfolding. For $v_1 = \cdots = v_k$, Montanari and Richard consider the unfolding $S = \{1, \ldots, \lfloor k/2 \rfloor\}$, estimating $\otimes^{\lfloor k/2 \rfloor} v_1$ by the first right singular vector of $\text{Mat}_S(X)$. They prove that $\otimes^{\lfloor k/2 \rfloor} v_1$ is (partially) recovered for $\lambda \gtrsim n_1^{\lceil k/2 \rceil/2}$, and conjecture sufficiency of $\lambda \gtrsim n_1^{k/4}$ (note that these bounds differ only for odd $k$). Theorem 5.8 of Hopkins et al. [15] completes this conjecture for $k = 3$, establishing sufficiency of $\lambda \geq (1 + \varepsilon)n_1^{3/4}$.

Complete analysis of unfolding—handling tensors of arbitrary order and asymmetric dimensions—relies on results in Section 1.3, which reveal (1) necessary and sufficient thresholds for recovery and (2) the exact limiting performance of estimates. In addition, spiked matrix results yield the exact limiting cosine similarity of unfolding procedures, not obtained in [20] or [15]. The recent disproportional results summarized in Lemma 1.2 are crucial as unfoldings such as $\text{Mat}_j(X)$, $j \in [k-1]$, have limiting aspect ratio zero (recall that $n_1 \leq \cdots \leq n_k$). Depending on the relative growth rates of $n_1, \ldots, n_k$, unfoldings such as $\text{Mat}_k(X)$ may fall under disproportional "tall" growth ($n_k^2/N_1 \to 0$), proportional growth ($n_k^2/N_1 \asymp 1$), or disproportional "wide" growth ($n_k^2/N_1 \to \infty$). Our analysis is similar to that of Ben Arous et al. [5], which permits arbitrary order $k$ yet assumes $n_1 = \cdots = n_k$.

**Theorem 2.1.** *Let $S \subset [k]$. If $N(S)^2/N_1 \to 0$, the first right singular vector $w$ of the unfolding $\text{Mat}_S(X)$ partially recovers $\otimes_{j \in S} v_j$ if and only if $\liminf \lambda/N_1^{1/4} > 1$. In particular, for $\lambda = \tau(1 + o(1))N_1^{1/4}$ with $\tau \geq 0$ fixed,*

$$|\langle \otimes_{j \in S} v_j, w \rangle|^2 \xrightarrow{a.s.} \overleftarrow{c}^2(\tau) . \qquad (12)$$

*If $N(S)^2/N_1 \asymp 1$, $\otimes_{j \in S} v_j$ is partially recovered if and only if $\liminf \lambda/N_1^{1/4} > 1$. For $\gamma_{n_1} \to \gamma$ and $\lambda = \tau(1 + o(1))N_1^{1/4}$,*

$$|\langle \otimes_{j \in S} v_j, w \rangle|^2 \xrightarrow{a.s.} c^2(\tau\gamma^{1/4}, \gamma) . \qquad (13)$$

*Finally, if $N(S)^2/N_1 \to \infty$, $\otimes_{j \in S} v_j$ is partially recovered if and only if $\liminf \lambda/\sqrt{N(S)} > 0$. For $\lambda = \tau(1 + o(1))\sqrt{N(S)}$,*

$$|\langle \otimes_{j \in S} v_j, w \rangle|^2 \xrightarrow{a.s.} \overrightarrow{c}^2(\tau) . \qquad (14)$$

All proofs are deferred to the supplement. As the recovery threshold is identical across all unfoldings $S$ such that $N(S)^2/N_1 \lesssim 1$, we propose the following simple algorithm to estimate $v_j$, which may be iterated for $j \in [k]$:

---

**Algorithm 1** Tensor unfolding

---

**Input:** $X, j$
**return** first right singular vector of $\mathrm{Mat}_j(X)$

---

**Corollary 2.1.1.** *Algorithm 1 partially recovers $v_1, \ldots, v_{k-1}$ if and only if $\liminf \lambda/N_1^{1/4} > 1$. Vector $v_k$, corresponding to the largest dimension, is partially recovered as well if $n_k^2/N_1 \lesssim 1$. On the other hand, if $n_k^2/N_1 \to \infty$, $v_k$ is recovered if and only if $\liminf \lambda/\sqrt{n_k} > 0$.*

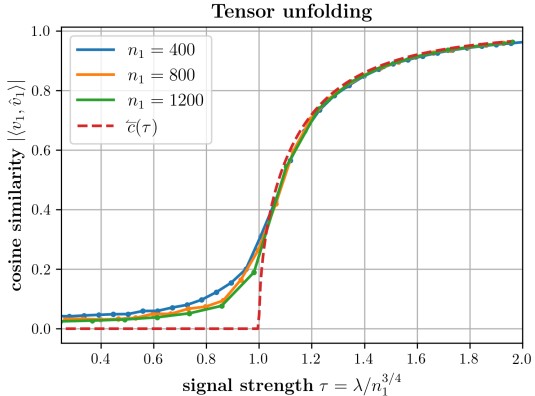

Figure 1: Simulations of tensor unfolding with $k = 3$, supersymmetric signal $v_1 = v_2 = v_3$, and $n_1 \in \{400, 800, 1200\}$. Solid lines display empirical cosine similarities (each point is the average of 50 realizations). The dashed line is the theoretical limit $\overleftarrow{c}(\tau)$, which agrees closely. Below the phase transition located at $\tau = 1$, $\hat{v}_1$ is approximately uniformly distributed on the surface of the unit sphere, so $|\langle v_1, \hat{v}_1 \rangle| = O(n^{-1/2})$.

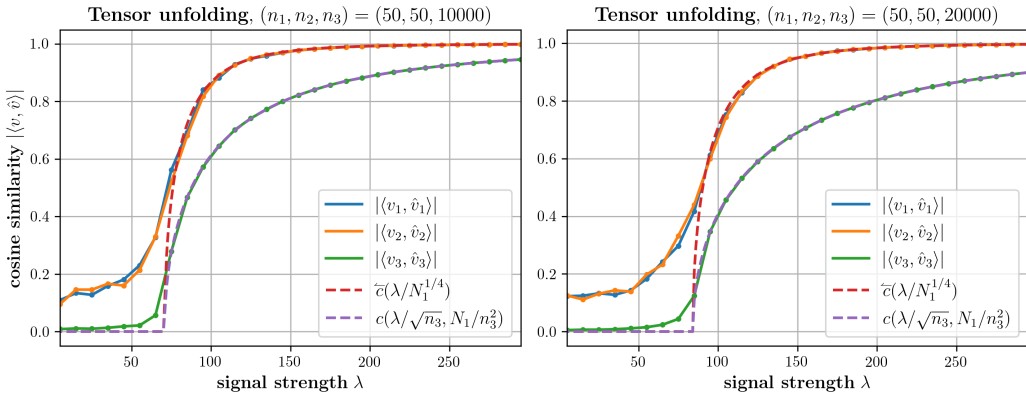

Figure 2: Simulations of tensor unfolding with $k = 3$, $n_1 = n_2 = 50$, and $n_3 = 10000$ (left) or $n_3 = 20000$ (right). Solid lines display empirical cosine similarities (each point is the average of 50 realizations). The dashed lines are theoretical limits based on Theorem 2.1, which agree closely. As $n_3/(n_1 n_2) = O(1)$, we compare $|\langle v_1, \hat{v}_1 \rangle|$ and $|\langle v_2, \hat{v}_2 \rangle|$ to $\overleftarrow{c}(\tau)$ and $|\langle v_3, \hat{v}_3 \rangle|$ to $c(\tau(N_1/n_3^2)^{1/4}, N_1/n_3^2) = c(\lambda/\sqrt{n_3}, N_1/n_3^2)$.

## 3 Partial Tracing

Hopkins et al. [16] propose the partial tracing method for tensors of order $k = 3$ as an alternative to unfolding. Their approach generalizes to orders $k \geq 3$, although it inherently relies on supersymmetry

of the signal, $v_1 = \cdots = v_k$. The partial tracing operator $\mathrm{Tr}_k : \otimes^k \mathbb{R}^n \mapsto \mathbb{R}^{n \times n}$ constructs a matrix by linearly combining tensor slices with partial trace weights:

$$\mathrm{Tr}_k(T) = \sum_{i_1,\ldots,i_{k-2} \in [n]} \mathrm{tr}(T_{i_1,\ldots,i_{k-2}}) T_{i_1,\ldots,i_{k-2}} \quad . \tag{15}$$

This operation of reducing the order of a tensor by linear combinations of slices is called contraction in tensor analysis.

---

**Algorithm 2** Partial tracing

---

**Input:** $X$
**return** first right singular vector of $\mathrm{Tr}_k(X)$

---

For $k = 3$, the runtime of Algorithm 2 is $O(n^3)$, while that of unfolding is $O(n^3 \log n)$ (see Table 3 of [16]). Hopkins et al. established a bound on the recovery threshold of partial tracing which is worse than that of unfolding by a logarithmic factor: $\lambda \gtrsim n^{3/4}(\log n)^{1/2}$. We eliminate the logarithmic factor here: while delivering improvements in runtime, Algorithm 2 is asymptotically equivalent to unfolding in recovery performance.

**Theorem 3.1.** *The first right singular vector $\hat{v}_1$ of the partial trace matrix $\mathrm{Tr}_k(X)$ partially recovers $v_1$ if and only if $\liminf \lambda/n_1^{k/4} > 1$. For $\lambda = \tau(1 + o(1))n_1^{k/4}$, where $\tau$ is fixed,*

$$|\langle v_1, \hat{v}_1 \rangle|^2 \xrightarrow{a.s.} \overleftarrow{c}^2(\tau) \, . \tag{16}$$

Under $n_1 = \cdots = n_k$, the behavior of unfolding is given by (12), matching (16) exactly. Thus, unfolding and partial tracing are asymptotically equivalent in performance.

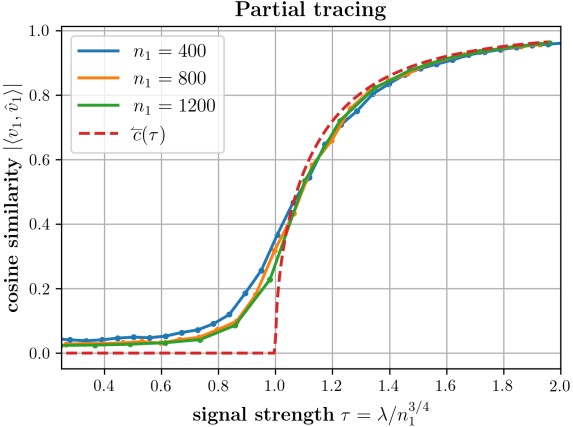

Figure 3: Simulations of partial tracing with $k = 3$, supersymmetric signal $v_1 = v_2 = v_3$, and $n_1 \in \{400, 800, 1200\}$. Solid lines display empirical cosine similarities (each point is the average of 50 realizations). The dashed line is the theoretical limit $\overleftarrow{c}(\tau)$, which agrees closely. Note the similarity to Figure 1.

## 4 Exact Recovery

Montanari and Richard [20] consider a two-step recursive unfolding procedure that remarkably *exactly* recovers $v_1$ above the threshold of unfolding, assuming even order $k$, hypercubical shape $n_1 = \cdots = n_k$, and a supersymmetric signal, $v_1 = \cdots = v_k$. In this section, we prove that exact recovery is possible for tensors of arbitrary axis lengths. We consider generalizations of the power iteration and recursive unfolding algorithms of [20] (initialized via tensor unfolding). Tensor power iteration is previously studied in [17] (assuming a supersymmetric signal) and [21] (assuming

$n_1 \asymp n_2 \asymp \cdots \asymp n_k$), though these works assume the initial iterate is independent of $X$, precluding initialization via unfolding.

In our more general setting, we find that unfolding-initialized power iteration achieves *exact* recovery above the *partial* recovery threshold of unfolding, established in Corollary 2.1.1 (assuming $n_k^2/N_1 \to 0$). Recursive unfolding exactly recovers $v_1, \ldots, v_{k-1}$ with no limitation on the growth rate of $n_k$. Power iteration, which requires initial "warm" estimates of $v_1, \ldots, v_k$, may fail when $n_k^2$ and $N_1$ are comparable. Recursive unfolding, on the other hand, does not rely upon an initial estimate of $v_k$ to recover $v_1, \ldots v_{k-1}$ exactly.

---

**Algorithm 3** Power iteration

> **Input:** $X$, initial estimates $\hat{v}_1, \ldots, \hat{v}_k$ from Algorithm 1
> Iterate until convergence:
> > **for** $j$ `from` 1 `to` $k$:
> > > $w_j \leftarrow X \times_1 \hat{v}_1 \cdots \times_{j-1} \hat{v}_{j-1} \times_{j+1} \hat{v}_{j+1} \cdots \times_k \hat{v}_k$
> > > $\hat{v}_j \leftarrow w_j/\|w_j\|_2$
> > **end for**
> **return** $\hat{v}_1, \ldots, \hat{v}_k$

---

We prove that exact recovery is achieved in a single iteration of the outer loop. In practice, iterating until estimates converge is beneficial.

**Theorem 4.1.** *If* $\liminf \lambda/N_1^{1/4} > 1$ *and* $n_k^2/N_1 \to 0$, *Algorithm 3 recovers* $v_1, \ldots, v_k$ *exactly: denoting by* $\hat{v}_1, \ldots, \hat{v}_k$ *the estimates after a single iteration of the outer loop,*

$$|\langle v_j, \hat{v}_j \rangle| \xrightarrow{a.s.} 1, \qquad j \in [k]. \tag{17}$$

*Estimating the signal strength* $\lambda$ *by* $\hat{\lambda} = \langle X, \hat{v}_1 \otimes \cdots \otimes \hat{v}_k \rangle$, *we have*

$$\lambda^{-1}\|\hat{\lambda}\hat{v}_1 \otimes \cdots \otimes \hat{v}_k - \lambda v_1 \otimes \cdots \otimes v_k\|_F \xrightarrow{a.s.} 0. \tag{18}$$

We next consider a recursive unfolding procedure. As stated above, recursive unfolding is guaranteed to recover $v_1, \ldots, v_{k-1}$ exactly if $\liminf \lambda/N_1^{1/4} > 1$; no bound is needed on the growth rate of $n_k$ as in Theorem 4.1. In practice, it is beneficial to iterate Algorithm 4.

---

**Algorithm 4** Recursive unfolding

> **Input:** $X$, initial estimates $\hat{v}_1, \ldots, \hat{v}_k$ from Algorithm 1
> **for** $j$ `from` 2 `to` $k$:
> > $\hat{v}_j \leftarrow$ first right singular vector of $\text{Mat}_{j-1}(X \times_1 \hat{v}_1)$
> **end for**
> $\hat{v}_1 \leftarrow$ first right singular vector of $\text{Mat}_1(X \times_2 \hat{v}_2)$
> **return** $\hat{v}_1, \ldots, \hat{v}_k$

---

**Theorem 4.2.** *Under* $\liminf \lambda/N_1^{1/4} > 1$, *Algorithm 4 recovers* $v_1, \ldots, v_{k-1}$ *exactly. Moreover,* $v_k$ *is recovered exactly if* $n_k^2/N_1 \lesssim 1$.

The analysis of Algorithm 4 entails careful application of spiked matrix model results. By the linearity of the unfolding operator,

$$\text{Mat}_{j-1}(X \times_1 \hat{v}_1) = \lambda\langle v_1, \hat{v}_1 \rangle(\otimes_{\ell \in [k]\setminus\{1,j\}} v_\ell)v_j^\top + \text{Mat}_{j-1}(Z \times_1 \hat{v}_1). \tag{19}$$

Observe that $Z \times_1 \hat{v}_1$ is a reshaping of the vector $\text{Mat}_1(Z)\hat{v}_1$. As $\hat{v}_1$ and $Z$ are dependent, the second term on the right-hand side is not a matrix of i.i.d. entries. Despite dependencies, we claim that appropriately scaled, $\text{Mat}_1(Z)\hat{v}_1$ is distributed as Gaussian noise, in which case exact recovery thresholds are a consequence of spiked matrix model results of Section 1.3 applied to (19).

Below, we display simulation results for power iteration and recursive unfolding and compare to tensor unfolding and partial tracing.

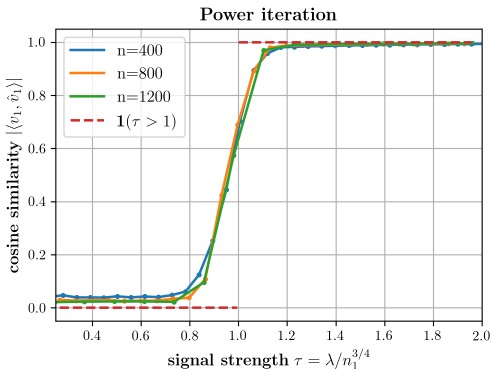

Figure 4: Simulations of power iteration with $k = 3$, supersymmetric signal $v_1 = v_2 = v_3$, and $n_1 \in \{400, 800, 1200\}$. Solid lines display empirical cosine similarities (each point is the average of 50 realizations). The dashed line is the theoretical prediction $\mathbf{1}(\tau > 1)$. Power iteration typically converges within five iterations.

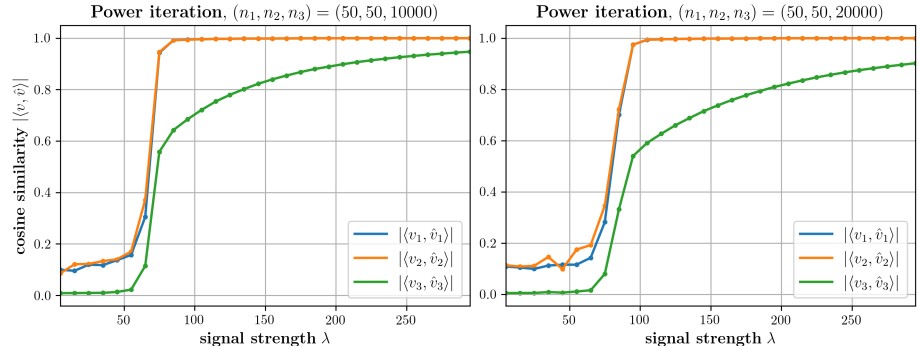

Figure 5: Simulations of power iteration with $k = 3$, $n_1 = n_2 = 50$, and $n_3 = 10000$ (left) or $n_3 = 20000$ (right). Solid lines display empirical cosine similarities (each point is the average of 50 realizations). Vector $v_3$, corresponding to the longest tensor axis, is estimated less well than $v_1, v_2$.

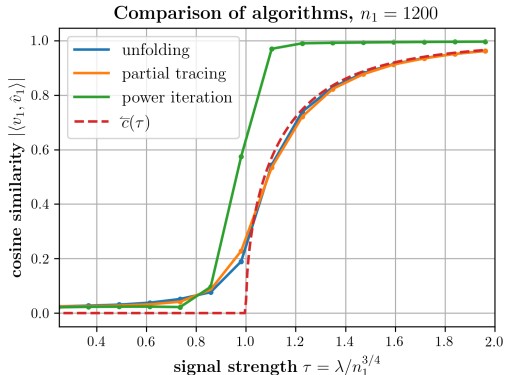

Figure 6: Simulations of a unfolding, partial tracing, and power iteration with $k = 3$, supersymmetric signal $v_1 = v_2 = v_3$, and $n_1 = 1200$. Solid lines display empirical cosine similarities (each point is the average of 50 realizations). The dashed line is the theoretical limit of unfolding and partial tracing, $\overleftarrow{c}(\tau)$. Near the phase transition located at $\tau = 1$, the cosine similarity of power iteration is over twice that of unfolding.

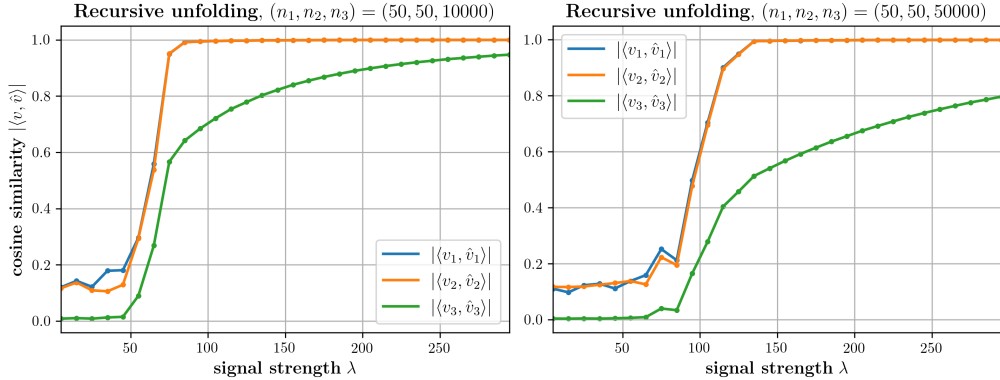

Figure 7: Simulations of recursive unfolding with $k = 3$, $n_1 = n_2 = 50$, and $n_3 = 10000$ (left) or $n_3 = 50000$ (right). Solid lines display empirical cosine similarities (each point is the average of 50 realizations). The performance of recursive unfolding is quite similar to that of power iteration.

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

## Appendix

*proof of Theorem (2.1).* By (11), $\text{Mat}_S(X)$ is a spiked matrix with aspect ratio $N(S)^2/N_1$ and signal strength $\lambda$ (as $\otimes_{j \in S} v_j$ is unit norm). There are four cases to consider:

- Let $N(S)^2/N_1 \to 0$; by the first point of (6), the recovery threshold lies at $(N_1/N(S) \cdot N(S))^{1/4} = N_1^{1/4}$.

- Let $N(S)^2/N_1 \to \gamma \in (0, 1]$; by the first point of (5), partial recovery occurs for $\liminf \lambda/\sqrt{N_1/N(S)} > \gamma^{1/4}$. Equivalently, $\liminf \lambda/N_1^{1/4} > 1$. Under $\lambda = \tau(1 + o(1))N_1^{1/4}$, $\lim \lambda/\sqrt{N_1/N(S)} = \tau\gamma^{1/4}$, implying $|\langle \otimes_{j \in S} v_j, w \rangle|^2 \xrightarrow{a.s.} c^2(\tau\gamma^{1/4}, \gamma)$.

- Let $N(S)^2/N_1 \to \gamma \in (1, \infty]$; by the second point of (5), partial recovery occurs for $\liminf \lambda/\sqrt{N(S)} > \gamma^{-1/4}$. Equivalently, $\liminf \lambda/N_1^{1/4} > 1$. Under $\lambda = \tau(1 + o(1))N_1^{1/4}$, $\lim \lambda/\sqrt{N(S)} = \tau\gamma^{-1/4}$, implying $|\langle \otimes_{j \in S} v_j, w \rangle|^2 \xrightarrow{a.s.} c^2(\tau\gamma^{-1/4} \cdot \gamma^{1/2}, \gamma) = c^2(\tau\gamma^{1/4}, \gamma)$.

- Let $N(S)^2/N_1 \to \infty$; by the second point of (7), the recovery threshold lies at $\sqrt{N(S)}$.

$\square$

*Proof of Theorem 3.1.* We provide proof for $k = 3$; the proof for higher orders is similar and omitted. For notational simplicity, we suppress the subscripts of $n_1$ and $v_1$.

Let $w_i = \text{tr}(Z_i)$, $i \in [n]$, and $w = (w_1, \ldots, w_n)^\top$. Expanding the partial trace matrix,

$$
\begin{aligned}
\text{Tr}_k(X) &= \sum_{i=1}^n (\lambda v_i + w_i)X_i = \sum_{i=1}^n \left[ (\lambda^2 v_i^2 + \lambda v_i w_i)vv^\top + (\lambda v_i + w_i)Z_i \right] \\
&= \sum_{i=1}^n \left[ (\lambda^2 v_i^2 + \lambda v_i w_i)vv^\top + (\lambda v_i + w_i)(Z_i - \text{diag}(Z_i - \widetilde{Z}_i)) + (\lambda v_i + w_i)\text{diag}(Z_i - \widetilde{Z}_i) \right],
\end{aligned}
$$

(20)

where $\widetilde{Z}$ is an independent copy of $Z$.

Let $M = \|\lambda v + w\|_2^{-1} \sum_{i=1}^{n} (\lambda v_i + w_i)(Z_i - \text{diag}(Z_i - \widetilde{Z}_i))$. As $w_i$ and $Z_i - \text{diag}(Z_i - \widetilde{Z}_i)$ are independent and the Gaussian distribution is rotationally invariant, $M \overset{d}{=} Z_1$. Hence, we have

$$\|\lambda v + w\|_2^{-1} \text{Tr}_k(X) = \alpha v v^\top + M + \|\lambda v + w\|_2^{-1} \sum_{i=1}^{n} (\lambda v_i + w_i) \text{diag}(Z_i - \widetilde{Z}_i), \quad (21)$$

where $\alpha = \|\lambda v + w\|_2^{-1}(\lambda^2 + \lambda \langle v, w \rangle)$. The partial trace matrix is thus proportional to a perturbation of a spiked matrix with aspect ratio $\gamma = 1$.

Let $\mathcal{E}$ denote the third term on the right-hand side of (21); we shall prove $n^{-1/2}\|\mathcal{E}\|_2 \xrightarrow{a.s.} 0$. Denoting $u = (\lambda v + w)/\|\lambda v + w\|_2$, the diagonal entries of $\sum_{i=1}^{n} u_i \text{diag}(\widetilde{Z}_i)$ are i.i.d. Gaussians with variance one, implying $\|\sum_{i=1}^{n} u_i \text{diag}(\widetilde{Z}_i)\|_2 \lesssim \sqrt{\log n}$, almost surely. Similarly, since $w \sim \mathcal{N}(0, nI_n)$ and $\|\lambda v + w\|_2 = \Theta_{a.s.}(\lambda + n)$,

$$\lambda \|\lambda v + w\|_2^{-1} \Big\| \sum_{i=1}^{n} v_i \text{diag}(Z_i) \Big\|_2 \lesssim \sqrt{\log n}\Big(1 + \frac{\lambda}{n}\Big). \quad (22)$$

To bound the remaining term of $\mathcal{E}$, let $\mathcal{Z} \in \mathbb{R}^{n \times n}$ denote the matrix with entries $\mathcal{Z}_{ij} = Z_{ijj}$, $i, j \in [n]$, in which case we may write

$$\Big\| \sum_{i=1}^{n} w_i \text{diag}(Z_i) \Big\|_2 = \sup_{1 \le j \le n} \Big| \sum_{i=1}^{n} w_i Z_{ijj} \Big| = \sup_{1 \le j \le n} |e_j^\top \mathcal{Z}^\top \mathcal{Z} \mathbf{1}_n|, \quad (23)$$

where $\mathbf{1}_n$ is the length-$n$ vector of ones. As $e_j^\top \mathcal{Z}^\top \mathcal{Z} e_j \sim \chi_n^2$ and

$$e_j^\top \mathcal{Z}^\top \mathcal{Z}(\mathbf{1}_n - e_j) \overset{d}{=} \sqrt{n-1} e_j^\top \mathcal{Z}^\top \widetilde{\mathcal{Z}} e_j \sim \sqrt{n-1} \cdot \mathcal{N}(0,1) \cdot \sqrt{\chi_n^2},$$

standard bounds such as (2.19) in [24] yield

$$\mathbf{P}\Big( \sup_{1 \le j \le n} |e_j^\top \mathcal{Z}^\top \mathcal{Z} \mathbf{1}_n - n| > c \cdot n \Big) \lesssim n e^{-Cn}, \quad (24)$$

where $c, C > 0$ are constants. As the right-hand side is summable, the Borel-Cantelli lemma implies

$$\sup_{1 \le j \le n} |e_j^\top \mathcal{Z}^\top \mathcal{Z} \mathbf{1}_n| \lesssim n, \quad (25)$$

almost surely. Collecting the above bounds, we have that $n^{-1/2}\|\mathcal{E}\|_2 \xrightarrow{a.s.} 0$.

By Weyl's inequality, the limiting spectral distribution of $n^{-1/2}(M + \mathcal{E})$ equals that of $n^{-1/2}M$, the quarter circle law. The limits (5) from Lemma 1.1 therefore apply to $\alpha v v^\top + M + \mathcal{E}$ as well (see [7]). Basic calculations yield (1) $\liminf \alpha/\sqrt{n} > 1$ if and only if $\liminf \lambda/n^{3/4} > 1$, and (2) under $\lambda = \tau(1 + o(1))n^{3/4}$, $\alpha = \tau^2(1 + o_{a.s.}(1))\sqrt{n}$. Therefore, $\hat{v}$ partially recovers $v$ if and only if $\liminf \lambda/n^{3/4} > 1$, and under $\lambda = \tau(1 + o(1))n^{3/4}$,

$$|\langle v, \hat{v} \rangle|^2 \xrightarrow{a.s.} c^2(\tau^2, 1) = \overleftrightarrow{c}^2(\tau), \quad (26)$$

completing the proof. $\qquad \square$

*Proof of Theorem 4.1.* By the linearity of the operators $\times_1, \ldots, \times_k$,

$$w_j = \lambda \prod_{\ell \in [k] \setminus \{j\}} \langle v_\ell, \hat{v}_\ell \rangle \cdot v_j + Z \times_1 \hat{v}_1 \cdots \times_{j-1} \hat{v}_{j-1} \times_{j+1} \hat{v}_{j+1} \cdots \times_k \hat{v}_k, \quad (27)$$

$$\langle v_j, w_j \rangle = \lambda \prod_{\ell \in [k] \setminus \{j\}} \langle v_\ell, \hat{v}_\ell \rangle + \big\langle Z, \hat{v}_1 \otimes \cdots \otimes \hat{v}_{j-1} \otimes v_j \otimes \hat{v}_{j+1} \cdots \otimes \hat{v}_k \big\rangle. \quad (28)$$

Assume that (17) holds for $\ell \in [j-1]$. For $\ell \in \{j+1, \ldots, k\}$, Corollary 2.1.1 implies $|\langle v_\ell, \hat{v}_\ell \rangle|$ is bounded away from zero. The first term on the right-hand side of (28) is therefore $\Theta_{a.s.}(\lambda)$. The second term is bounded by the spectral norm of $Z$: using Theorem 1 of [23],

$$\big\langle Z, \hat{v}_1 \otimes \cdots \otimes \hat{v}_{j-1} \otimes v_j \otimes \hat{v}_{j+1} \otimes \cdots \otimes \hat{v}_k \big\rangle \le \sup_{u_j \in \mathbb{S}^{j-1}, j \in [k]} \langle Z, u_1 \otimes \cdots \otimes u_k \rangle \lesssim \sqrt{n_k}, \quad (29)$$

almost surely. Thus, as $(\sqrt{n_k}/\lambda)^4 \lesssim n_k^2/N_1 \to 0$,

$$|\langle v_j, \hat{v}_j \rangle| = 1 + O_{a.s.}\left(\frac{\sqrt{n_k}}{\lambda}\right) \xrightarrow{a.s.} 1, \tag{30}$$

from which (17) follows inductively.

Equation (18) follows from (17) and (29), which imply $\|v_1 \otimes \cdots \otimes v_k - \hat{v}_1 \otimes \cdots \otimes \hat{v}_k\|_F \xrightarrow{a.s.} 0$ and

$$\lambda^{-1}|\hat{\lambda}| = \left| \prod_{j=1}^k \langle v_j, \hat{v}_j \rangle + \lambda^{-1} \langle Z, \hat{v}_1 \otimes \cdots \hat{v}_k \rangle \right| \xrightarrow{a.s.} 1. \tag{31}$$

$\square$

*Proof of Theorem 4.2.* We shall prove $v_2, \ldots, v_k$ are recovered exactly; proofs for the first and last axes are similar and omitted. By the linearity of the unfolding operator,

$$\mathrm{Mat}_{j-1}(X \times_1 \hat{v}_1) = \lambda \langle v_1, \hat{v}_1 \rangle (\otimes_{\ell \in [k] \setminus \{1,j\}} v_\ell) v_j^\top + \mathrm{Mat}_{j-1}(Z \times_1 \hat{v}_1). \tag{32}$$

Observe that $Z \times_1 \hat{v}_1$ is a reshaping of the vector $\mathrm{Mat}_1(Z)\hat{v}_1$. As $\hat{v}_1$ and $Z$ are dependent, the second term on the right-hand side is not a matrix of i.i.d. entries. Despite dependencies, we claim that appropriately scaled, $\mathrm{Mat}_1(Z)\hat{v}_1$ is Gaussian noise, in which case exact recovery thresholds are a consequence of spiked matrix model results of Section 1.3 applied to (32).

By definition, $\hat{v}_1$ is the first eigenvector of the symmetric matrix

$$\mathrm{Mat}_1(X)^\top \mathrm{Mat}_1(X) = \lambda^2 v_1 v_1^\top + \mathrm{Mat}_1(Z)^\top \mathrm{Mat}_1(Z) + \mathcal{E}, \tag{33}$$

where $\mathcal{E}$ is a rank-two matrix given by

$$\mathcal{E} = \lambda v_1 (\otimes_{\ell \in [k] \setminus \{1\}} v_\ell)^\top \mathrm{Mat}_1(Z) + \lambda \mathrm{Mat}_1(Z)^\top (\otimes_{\ell \in [k] \setminus \{1\}} v_\ell) v_1^\top.$$

Let $\tilde{v}_1$ denote the first eigenvector of $\mathrm{Mat}_1(X)^\top \mathrm{Mat}_1(X) - \mathcal{E}$; without loss of generality, we assume $\hat{v}_1^\top \tilde{v}_1 \geq 0$. Since $\mathrm{Mat}_1(Z)^\top (\otimes_{\ell \in [k] \setminus \{1\}} v_\ell) \sim \mathcal{N}(0, I_{n_1})$, we have $\|\mathcal{E}\|_2 \lesssim \lambda \sqrt{n_1}$ almost surely. By Theorem 1.1 of [13], the spectral gap of $\mathrm{Mat}_1(X)^\top \mathrm{Mat}_1(X)$ is $\Theta(\lambda^2)$. Thus, using the Davis-Kahan theorem (see Corollary 3 of [25]), we have

$$\|\hat{v}_1 - \tilde{v}_1\|_2 \lesssim \frac{\|\mathcal{E}\|_2}{\lambda^2} \xrightarrow{a.s.} 0. \tag{34}$$

Let $\mathrm{Mat}_1(Z) = U\Lambda V^\top$ be a singular value decomposition. As $Z$ contains i.i.d. Gaussian entries, (1) $U$, $\Lambda$, and $V$ are independent, (2) $U$ and $V$ are Haar-distributed. Moreover, as $\tilde{v}_1$ is the first eigenvector of $\lambda^2 v_1 v_1^\top + \mathrm{Mat}_1(Z)^\top \mathrm{Mat}_1(Z) = \lambda^2 v_1 v_1^\top + V\Lambda^2 V^\top$, $\tilde{v}_1$ is independent of $U$. Thus, $U\Lambda V^\top \tilde{v}_1 / \|\Lambda V^\top \tilde{v}_1\|_2$ is uniform on $\mathbb{S}^{N_2 - 1}$. Generating $\xi \sim \chi_{N_2}^2$ independent of $Z$, it follows that

$$\frac{\xi}{\sqrt{N_2}} \cdot \frac{\mathrm{Mat}_1(Z)\tilde{v}_1}{\|\Lambda V^\top \tilde{v}_1\|_2} \sim \mathcal{N}(0, I_{N_2}). \tag{35}$$

Defining the constant $\alpha = \xi/(\sqrt{N_2}\|\Lambda V^\top \tilde{v}_1\|_2)$, we deduce that $\alpha Z \times_1 \tilde{v}_1$ (which is a reshaping of the left-hand side of (35)) is distributed as a tensor with i.i.d. Gaussian entries—despite dependencies between $Z$ and $\tilde{v}_1$. Additionally, since $\Lambda_{11}/\sqrt{N_2} \xrightarrow{a.s.} 1$ and $\Lambda_{n_1 n_1}/\sqrt{N_2} \xrightarrow{a.s.} 1$, we have $\|\Lambda V^\top \tilde{v}_1\|_2/\sqrt{N_2} \xrightarrow{a.s.} 1$ and $\alpha \xrightarrow{a.s.} 1$.

Thus, we conclude that

$$\alpha \mathrm{Mat}_{j-1}(X \times_1 \tilde{v}_1) = \lambda(1 + o_{a.s.}(1)) \langle v_1, \tilde{v}_1 \rangle (\otimes_{\ell \in [k] \setminus \{1,j\}} v_\ell) v_j^\top + \widetilde{Z}, \quad j \in \{2, \ldots, k-1\}, \tag{36}$$

where $\widetilde{Z} \in \mathbb{R}^{n_j \times (N_2/n_j)}$ contains i.i.d. Gaussian entries, enabling us to apply spiked matrix results. Let $\tilde{v}_j$ denote the first right singular vector of $\mathrm{Mat}_{j-1}(X \times_1 \tilde{v}_1)$ (equivalently, that of $\alpha \mathrm{Mat}_{j-1}(X \times_1 \tilde{v}_1)$); without loss of generality, we assume $\hat{v}_1^\top \tilde{v}_1 \geq 0$. In particular, since $|\langle v_1, \tilde{v}_1 \rangle| \asymp |\langle v_1, \hat{v}_1 \rangle| \asymp 1$ by Corollary 2.1.1 and (34), Lemma 1.2 implies $|\langle v_j, \tilde{v}_j \rangle| \xrightarrow{a.s.} 1$ (we have $\lambda \gg (n_j \cdot N_2/n_j)^{1/4}$).

It therefore suffices to prove that $\|\hat{v}_j - \tilde{v}_j\|_2 \xrightarrow{a.s.} 0$. By Theorem 2.3 of [5] or Theorem 1.1 of [13] and Cauchy's interlacing inequality, the spectral gap of $\mathrm{Mat}_{j-1}(X \times_1 \tilde{v}_1)^\top \mathrm{Mat}_{j-1}(X \times_1 \tilde{v}_1)$ is $\Theta(\lambda^2)$. Let $\mathcal{Z}$ denote the reshaping of $Z$ with dimensions $n_1 \times n_j \times N_2/n_j$ and slices $\mathrm{Mat}_{j-1}(Z_i)$, $i \in [n_1]$. Using the bound

$$\|\mathrm{Mat}_{j-1}(Z \times_1 (\hat{v}_1 - \tilde{v}_1))\|_2 \leq \sup_{\substack{u_1 \in \mathbb{S}^{n_1-1}, u_2 \in \mathbb{S}^{n_j-1}, \\ u_3 \in \mathbb{S}^{N_2/n_j-1}}} \left(\mathcal{Z} \times_1 u_1 \times_2 u_2 \times_3 u_3\right) \cdot \|\hat{v}_1 - \tilde{v}_1\|_2$$

and Theorem 1 of [23],

$$\|\mathrm{Mat}_{j-1}(X \times_1 (\hat{v}_1 - \tilde{v}_1))\|_2 \leq \lambda\|\hat{v}_1 - \tilde{v}_1\|_2\|\mathrm{Mat}_{j-1}(v_2 \otimes \cdots \otimes v_k)\|_2 + \|\mathrm{Mat}_{j-1}(Z \times_1 (\hat{v}_1 - \tilde{v}_1))\|_2$$
$$\lesssim \left(\lambda + (N_2/n_j)^{1/2}\right)\|\hat{v}_1 - \tilde{v}_1\|_2,$$

(37)

almost surely. Thus, using the Davis-Kahan theorem (Theorem 4 of [25]), (34), and (37), we have

$$\|\hat{v}_j - \tilde{v}_j\|_2 \lesssim \frac{\lambda\left(\lambda + (N_2/n_j)^{1/2}\right)\|\hat{v}_1 - \tilde{v}_1\|_2}{\lambda^2} \xrightarrow{a.s.} 0,$$

(38)

completing the proof. $\square$

