# OpenReview forum: "Sharp Recovery Thresholds of Tensor PCA Spectral Algorithms"
_NeurIPS.cc/2023/Conference — NeurIPS 2023 poster_

### Official Review · Reviewer_tdbZ · 2023-06-20

**Soundness:** 4 excellent
**Presentation:** 4 excellent
**Contribution:** 1 poor
**Rating:** 3
**Confidence:** 5

**Summary:**

This paper considers the (Gaussian, nonsymmetric, rank-1) spiked tensor model introduced in Montanari and Richard.  It gives sharp thresholds for various matricization techniques (unfolding, partial traces, successive contraction).  The statements about random tensors are reduced to known results about left and right singular vectors of spiked non-symmetric random matrices.  The proofs of the new statements are a few lines long.  Some numerical results back up the performance statement about successive contraction.






**Strengths:**

The presentation is generally clear and concise.  The proofs seem reasonably detailed given the content matter.  Some of the results appear novel and improve over substantially longer published work.  Highlights include
1) Improving an analysis in Hopkins et al.
2) Generalizing the successive contraction method of Montanari and Richard with excellent performance properties.

**Weaknesses:**

The new theoretical content in this paper is very slight: the random matrix theory of spiked Gaussian matrices is extremely well-developed. Once it is noticed that the unfolded tensors satisfy the hypotheses of these theorems, the proof is immediate.

There is no further elaboration beyond this:
1) the paper does not show how this fits into a larger picture of non-random-matrix-theory approaches.
2) the paper does not attempt to expand interesting unexplored aspects which don't follow from textbook random matrix theory (the best hint of this are the numerics that appear for tau=1 at the end).
3) the new technique (in the abstract "Finally, we introduce an iterative algorithm based on successive contractions of the tensor") is rather a generalization of a technique of Montanari and Richard.  The analysis given is in the phase of perfect recovery, exploiting some of the specific structure of the spiked model.

This is a reasonable topic for a purely theoretical NeurIPS contribution, but there is not enough new material here to warrant publication.

**Questions:**

No questions.

**Limitations:**

See above.

---

### Official Review · Reviewer_mafC · 2023-06-22

**Soundness:** 3 good
**Presentation:** 3 good
**Contribution:** 2 fair
**Rating:** 6
**Confidence:** 4

**Summary:**

This paper studies recovery of planted signals from noisy tensors.
The model is the prototypical tensor PCA and the emphasis is that (i) the planted low rank tensors (in particular their dimensions) can be different along different modes; (ii) the joint scaling of different dimensions can deviate from the classical proportional regime.
The paper studies a few old and new natural algorithms and obtain sharp performance guarantees (specifically the overlap) for almost all of them.

**Strengths:**

Both the techniques and the results of this paper are neat.
It properly combines recent (and some not-so-recent) results in random matrix theory (RMT) and high-dimensional statistics and extend them to a satisfactory extent.
Once some prior results are properly organized, the proofs of the present results become rather straightforward.
But for ICML, I think this is enough.
Also, I thought about a subset of the questions studied in this paper before (perhaps before some of the recent RMT results were available) and at that time it was not clear how these results can be proved.

**Weaknesses:**

One important aspect of tensor PCA that the authors seem to ignore by large is the potential existence of information-computation gap.
The authors may want to check papers of Sam Hopkins, David Steurer, Andrea Montanari, Guy Bresler, Alex Wein (listed in random order) and many others.
They provide evidence of average case hardness from the perspectives of SoS, low-degree polynomials, reduction, statistical query lower bounds, AMP, statistical physics, etc.
I know the information-theoretic aspect and the info-comp gap is not the focus of this paper.
But when it comes to such a prototypical problem of tensor PCA, it's always good to say a few words about it so that the present results are properly positioned in the literature.



**Questions:**

1. In equation (1.3), should (r) in the superscript be (i)?

2. Line 58 'Where we estimate v_j^{(1)} by the first right singular vector...'. Here by 'first', I suppose lambda1 >= lambda2 >= ... is assumed. Am I right?

3. The set {v_j(k)} is assumed to be orthonormal. How strong is this assumption? It seems a bit more realistic to assume v_j^{(k)}'s are in general position. Also, my impression is that this will create quite some technical challenges. Correct me if I'm wrong.

3. Some theorem/lemma statements are not sufficiently clear. E.g., Theorem 2.1, line 120, 122, does 'recovered' still refer to partial recovery as in line 118? I suggest state it more formally, at least in the theorem statement. Please check thoroughly for similar issues.

4. Lemma 3.2 is the standard Gaussian conditioning lemma. It'll be nice to put a reference there. The standard one might be Lemma 11 of Bayati--Montanari (https://arxiv.org/abs/1001.3448), though there might be better ones. Please check the literature.

5. Proof of Theorem 3.1, I feel it's actually better to do the proof for general k instead of k=3, maybe in the full version of the paper if the authors wish.

6. The successive contraction algorithm is indeed quite natural. It reminds me of something called decimation AMP studied here https://arxiv.org/abs/2306.01412. That's in the context of *high*-rank matrix estimation (using AMP, but never mind that) and it estimates each component of the planted matrix sequentially and subtract it out for the next one.

7. I'm not sure if the authors are aware of the following line of work that analyzes MLE (which is computationally expensive) for tensor PCA using a novel technique. The results are not completely new since MLE has been well-understood and moreover their results require the assumption that the limit of overlap exists. However, their techniques involves standard RMT and seem conceptually more accessible. The authors may want to look into or even consider citing some of them.
- https://arxiv.org/abs/2108.00774 (square case)
- https://arxiv.org/abs/2112.12348 (rectangular case)
- and related works by roughly the same authors
- https://arxiv.org/abs/2110.01652 (spectra of contracted tensors)


Minor things:
- I found it slightly weird that the citations are in the format (1), (2), etc. Note that the same format is also used for bullet points (see line 95, 110). Very similar notation is also used for equation numbers (1.1), (1.2). To avoid confusion, I suggest use the more standard [1], [2] or [ABC20], [XYZ99] format for citation.

- Sometimes the first letter of 'proof of Theorem XXX' is not capitalized.

---

> ### Author Rebuttal · Authors · 2023-08-09
>
> Thank you for the detailed comments and questions! We are familiar with the information-computation gap for tensor PCA and agree with your suggestion that is should be mentioned in the introduction. Here are point-by-point replies:
>
> 1.	Yes, that is a typo.
>
> 2.	Yes, we will add a clarification.
>
> 3.	It is not an inconsequential assumption, although it is a generalization to most of the tensor PCA literature which assumes r=1. “Low rank” does not generalize from matrices to tensors as nicely. For example, tensor rank is often defined to be the minimal $r$ such that $T = \sum_{j=1}^r v_1^{(j)} \otimes \cdots \otimes v_k^{(j)}$, where $\{v_1^{(j)}, \ldots v_k^{(j)}\}$ are not necessarily orthogonal. A tensor that is low-rank under this definition may not have a decomposition into an equal number of terms with orthogonal vectors (this decomposition does always exist in the matrix case $k=2$).
>
> 4.	Yes, this is partial recovery. We will clarify this.
>
> 5.	Yes, this is standard. Thank you for the suggested reference, we have added it.
>
> 6.	We elected to write out the proof for $k=3$ for sake of simplicity, the general proof is nearly identical, although sums such as (3.2) become $k$ sums over each tensor axis.

---

### Official Review · Reviewer_PnQP · 2023-07-03

**Soundness:** 3 good
**Presentation:** 3 good
**Contribution:** 2 fair
**Rating:** 4
**Confidence:** 4

**Summary:**

This paper is concerned of the tensor PCA problem (low-rank tensor recovery with Gaussian noise), in which the authors proposed three new ways to approach the problem with theoretical guarantees.



**Strengths:**

1. This paper uses a succinct way to introduce to readers 3 different approaches of tensor PCA, all with formal guarantees. Therefore, a reader can quickly identify the approach they personally need.

2. All of the results are based on well-established matrix PCA results, so it is easier for readers to understand, and also gives people stronger confidence to use, as tensor analysis is notoriously hard.

**Weaknesses:**

1. Although the results themselves are fine, and solid to my eyes, but none of them provide new insights to the problem, nor did the authors come up with new or innovative techniques. It just seems to me that all this paper is doing is adapting a few matrix PCA techniques to the tensor regime, and only perform minimal changes to the procedure so that the framework can be used. This does not mean the results themselves are inferior, but the merit this paper brings is very limited beyond the verbatim theorems.

2. The paper presented three different approaches, but did not compare/contrast the three, so unfamiliar readers may not know which is the best technique to apply.

3. The simulation is limited (however, this is mostly a theoretical paper, so acceptable).

**Questions:**

N/A

---

> ### Author Rebuttal · Authors · 2023-08-09
>
> Thank you for your review of our manuscript. Tensor unfolding and partial trace, we discover, are asymptotically equivalent in performance. Successive contraction is shown in Section 4 to achieve exact recovery above the common threshold of tensor unfolding/partial trace, and we recommend it be used in practice. The benefits of successive contraction over tensor unfolding/partial trace in simulations is shown in Figure 2. We have conducted additional simulations in the case where $n_1 = n_2 \ll n_3$; please see the plots attached to our response.

---

> > ### Comment · Reviewer_PnQP · 2023-08-16
> >
> > Thank you for your response, and I acknowledge your arguments.
> >
> > I'm not saying the results themselves are not good, however, in my eyes they do not bring any new high-level ideas to the table, therefore probably better fitted for a journal submission where the authors can really flesh out the details.
> >
> > After deliberation I am keeping my original scores, and I thank the authors for their interactions again.

---

### Official Review · Reviewer_K1MF · 2023-07-04

**Soundness:** 2 fair
**Presentation:** 2 fair
**Contribution:** 2 fair
**Rating:** 4
**Confidence:** 4

**Summary:**

This paper studies tensor principal component analysis (PCA) in a high-dimensional asymptotic framework, where each array dimension tends to infinity. The authors analyze matricization-based approaches, which convert tensors to matrices and then apply spectral methods. They fully analyze tensor unfolding or reshaping and the partial trace approach, discovering their asymptotic equivalence in performance. They identify sharp thresholds in signal-to-noise ratio above which these algorithms partially recover the signal and provide exact formulas for their limiting performance. Finally, they introduce an iterative algorithm based on successive contractions of the tensor, which achieves exact signal recovery above the same thresholds characterizing partial recovery of unfolding and partial trace. The authors' approach relies on fundamental random matrix theory results.

**Strengths:**

The main strengths of the paper are:
1. The paper provides a comprehensive analysis of tensor PCA algorithms in a high-dimensional asymptotic framework, allowing for tensors of diverse dimensions and unrelated signals.
2. The paper introduces an iterative algorithm based on successive contractions of the tensor, which achieves exact signal recovery above the best-known algorithmic threshold.
3. The paper provides exact formulas for the limiting performance of the algorithms and identifies sharp thresholds in the signal-to-noise ratio above which these algorithms partially recover the signal.

**Weaknesses:**

The main weaknesses of the paper are:
1. The presented successive contraction is demonstrated to achieve exact recovery only asymptotically, while a non-asymptotic result would be more interesting to investigate. In fact, the authors should compare the obtained performance of the successive contraction method in a non-asymptotic regime with the MLE which was analyzed in [1].
2. It is unclear how the successive contraction method differs from the tensor power iteration. In fact, [2] showed that the latter achieves also exact recovery above the algorithmic threshold when initialized with signal estimates using the tensor unfolding method.

[1] Mohamed El Amine Seddik, Maxime Guillaud, and Romain Couillet. “When Random Tensors meet Random Matrices”. In: arXiv preprint arXiv:2112.12348 (2021)

[2] Arnab Auddy and Ming Yuan. “On Estimating Rank-One Spiked Tensors in the Presence of Heavy Tailed Errors”. In: arXiv preprint arXiv:2107.09660 (2021)

**Questions:**

Please refer to the weaknesses.

**Limitations:**

The main limitations of the paper are:
1. The paper considers an asymptotic analysis for Tensor PCA assuming the dimension of the tensor to grow to infinity, while a non-asymptotic analysis or rather a quantification of the fluctuations would be interesting to investigate.
2. Most of the presented results in the paper are relatively known in the literature (Tensor Unfolding and Partial Trace).
3. Lack of comparison of the presented successive contraction method with the tensor power iteration algorithm.

---

### Author Rebuttal · Authors · 2023-08-09

We thank the reviewers for the time and effort they have invested into the review of our manuscript, and for their helpful comments and suggestions. We would like to address the reviewers’ main criticism, that our results follow easily from PCA results of random matrix theory (RMT). The primary purpose of this work is to exhibit spiked matrix results (those of [2] as well as newer results of [1] and [3]) and demonstrate how they can be applied to machine learning problems. While well-known within RMT and to the reviewers, the machine learning community is less familiar (in our opinion) with this literature. There are numerous potential applications of this theory to machine learning, many of which we believe are underexplored. As machine learning problems often concern the setting where dimensions exceed sample size, the recent results of [1] and [3] (which generalize previous “proportional” results of [2] to the “disproportional” setting) may be particularly useful.

Our work concisely demonstrates the advantages of this RMT-backed approach in the context of tensor PCA. Indeed, application of spiked matrix results yields simple and elegant theorems which may be easily read by theoreticians as well as practitioners. For example, analysis of the partial trace method, originally done by challenging matrix concentration and perturbation calculations, is reduced to a few careful lines. Though our results are admittedly purely asymptotic, there are no hidden constants or logarithmic factors as in [4] (which frequently give practitioners pause). Even for modest-sized tensors, our simulations demonstrate close agreement with theory.
 Moreover, this approach reveals the asymptotic exact equivalence in performance of tensor unfolding and partial trace. To the best of our knowledge, we are the first to make this discovery. Our paper makes clear that for a number of (seemingly different) algorithms, performance fundamentally derives from spectral properties of spiked matrices.

We agree that what we have called successive contraction is simply a generalization of the power iteration algorithm of [5] and ought to be called as such. While previous analysis in [6] is limited to the setting where $n_1 \asymp n_2 \asymp \cdots \asymp n_k$, we make no assumptions on the relative rates of growth of $n_1, n_2, \ldots, n_k$. We have conducted additional simulations where $n_1 = n_2 = 50$ and $n_3 = 10000$ or $n_3 = 20000$, again demonstrating agreement with theory.
Please see the attached plots.

If accepted, we will clarify these questions in the revision.


[1] G. Ben Arous, D. Huang, and J. Huang. Long Random Matrices and Tensor Unfolding. arXiv 208 preprint arXiv2110.1021, 2021.

[2] F. Benaych-Georges and R. Rao Nadakuditi. The singular values and vectors of low rank perturbations of large rectangular random matrices. Journal of Multivariate Analysis, 111:120–135, 2012.

[3] M. Feldman. Spiked Singular Values and Vectors under Extreme Aspect Ratios. Journal of Multivariate Analysis, 196, 2023.

 [4] S. Hopkins, T. Schramm, J. Shi, and D. Steurer. Fast spectral algorithms from sum-of-squares proofs: tensor decomposition and planted sparse vectors. Proceedings of the 48th Annual ACM SIGACT Symposium on Theory of Computing, 178–191, 2016.

[5] A. Montanari and E. Richard. A statistical model for tensor PCA. Proceedings of the 27th International Conference on Neural Information Processing Systems, 2:2897–2905, 2014.

[6] M. E. A. Seddik, M. Guillaud, and R. Couillet. When random tensors meet random matrices. arXiv preprint arXiv:2112.12348, 2021.

---

### Decision · Program_Chairs · 2023-09-21

**Decision:**

Accept (poster)

**Comment:**

The reviewers have conducted a detailed reading of the results and comparison with the existing literature. At least one of the reviewer, tdbZ, states they are an expert in the area. The overall conclusion is that the results of the manuscript are fine, but follow overly directly from the exiting literature. Specifically that the proofs are extremely simple and in effect follow from noting that the variant of the question satisfies conditions of theorem from which the new results then follow. That said, it is agreed that the result are worthy of publishing, especially with the more novel result on partial trace and exact recovery in sections 3 and 4. The reviewers also agree that the manuscript is generally well written. The only issue is if the manuscript is suitable for NeurIPS or if it should be in a journal setting; summarised as "too little for NeurIPS." The authors and reviewers have engaged productively.  The manuscript has substantial new contributions with the non-proportional setting.  The contributions are mathematically rigorous.  The only issue for debate is if the manuscript is sufficiently novel for NeurIPS; and for this question my view is that in total yes, the contributions are sufficient for publication at NeurIPS, though best suited for a poster.